# Pivmecillinam for Uncomplicated Lower Urinary Tract Infections Caused by *Staphylococcus saprophyticus*—Cumulative Observational Data from Four Recent Clinical Studies

**DOI:** 10.3390/antibiotics8020057

**Published:** 2019-05-07

**Authors:** Filip Jansåker, Marianne Bollestad, Ingvild Vik, Morten Lindbæk, Lars Bjerrum, Niels Frimodt-Møller, Jenny Dahl Knudsen

**Affiliations:** 1Department of Clinical Microbiology, Hvidovre Hospital, University of Copenhagen, 2650 Hvidovre, Denmark; Inge.Jenny.Dahl.Knudsen@regionh.dk; 2Department of Clinical Microbiology, Rigshospitalet, University of Copenhagen, 2100 Copenhagen, Denmark; Niels.Frimodt-Moeller@regionh.dk; 3Division of Medicine, Stavanger University Hospital, 4068 Stavanger, Norway; mbollestad@hotmail.com; 4Antibiotic Centre of Primary Care, Department of General Practice, Institute of Health and Society, University of Oslo, 0318 Oslo, Norway; ingvild.vik@medisin.uio.no (I.V.); morten.lindbak@medisin.uio.no (M.L.); 5Section of General Practice and Research Unit of General Practice, Department of Public Health, University of Copenhagen, 1014 Copenhagen, Denmark; lbjerrum@sund.ku.dk

**Keywords:** Mecillinam, pivmecillinam, cystitis, UTI, *Staphylococcus saprophyticus*, mecillinam breakpoints

## Abstract

*Objectives*: To investigate pivmecillinam´s efficacy in uncomplicated lower urinary tract infection (UTI) caused by *Staphylococcus saprophyticus—*considered non-susceptible to mecillinam. *Methods*: Participants with confirmed UTIs caused by *S. saprophyticus* from four randomized controlled trials, where pivmecillinam was empirically given to females with symptoms of UTIs. The primary outcome was defined as a cumulative clinical effect—symptom resolution during the first eight days of therapy, without a recurrence of UTI symptoms in the long-term follow-up (approximately four weeks). Secondary outcomes included the bacteriological effect—elimination of the causative agent, with or without new uropathogenic bacteria present in the first control urine sample. Significant bacteriuria was defined as ≥10^3^ bacteria/mL. The antibiotic susceptibility testing was done by disc diffusion methodology, according to the European Committee on Antimicrobial Susceptibility Testing (EUCAST). *Results*: Seventy-four females (18–55 years) were empirically treated with pivmecillinam for UTIs caused by *S. saphrophyticus* (mean age 25 years; standard deviation (SD) 5.8). The cumulative clinical effect was 53/74 (72%), and the bacteriological effect was 51/59 (86%). *Conclusion*: Pivmecillinam showed a high clinical and bacteriological effect in UTIs caused by *S. saprophyticus* in these four clinical trials. The characterization of non-susceptibility for mecillinam regarding the treatment of UTIs caused by this common pathogen may need to be revised.

## 1. Introduction

Pivmecillinam (amdinocillin pivoxil), the oral prodrug of the antimicrobial active agent mecillinam (amdinocillin), is a bona fide agent for uncomplicated lower urinary tract infections (UTIs), because of its selectivity and effect against several species of Enterobacteriaceae. It is especially effective against *Escherichia coli,* the most common UTI pathogen, isolated in above 80% of UTI cases. The mode of action is also unique among the β-lactam antimicrobials, as mecillinam has a high specificity for the penicillin-binding protein-2 [1,2,3]. The bioavailability for pivmecillinam is high, and pharmacokinetic studies have shown the urine mecillinam concentration to exceed 200 mg/L [4,5]. The drug was introduced several decades ago into Scandinavian markets. Its use has increased significantly in recent years, because of the rise in resistance to other commonly used antimicrobials for UTIs [6,7,8], and its clinical effect against extended-spectrum β-lactamase (ESBL) producing *E. coli* [9,10]. Pivmecillinam has been used in Scandinavia for decades, with stable and low rates of resistance in *E. coli*, minor rates of collateral damage, and a low risk of clonal spread of resistance [6,7,8,11,12,13,14,15,16]. Adverse reactions most commonly include mild gastrointestinal symptoms [2,3]. Pivmecillinam also provides potential use for uncomplicated pyelonephritis, when resistance is high to other alternatives [17].

Although pivmecillinams efficacy for UTIs caused by *E. coli* is examined thoroughly, the efficacy for the second most common UTI pathogen—*Staphylococcus saprophyticus,* which comprises up to 10% of all UTIs [3]—is not well studied. This Gram-positive bacterium is considered in vitro resistant to mecillinam, with minimum inhibitory concentrations (MICs) of ≥8 mg/L, but there are no defined epidemiological cut-off values (ECOFFs) in the European Committee on Antimicrobial Susceptibility Testing (EUCAST) [18]. Nevertheless, the bacteriological effect of pivmecillinam for UTIs caused by *S. saprophyticus* has been reported with a similar effect to other oral antimicrobials [3]. Monsen et al. [19] previously described no correlation between the resistance to mecillinam and the bacteriological or clinical outcome (sub-analysis from a study by Ferry et al.) [20]. The authors also described a high clinical and bacteriological effect of pivmecillinam for UTIs caused by *S. saprophyticus,* with a significant superiority to the placebo.

In this study, we present (to our knowledge) the largest cumulative data from the clinical trials on pivmecillinam´s efficacy for UTIs caused by *S. saprophyticus.* The data is cumulated from four recent randomized controlled trials on the efficacy of empirical pivmecillinam for UTIs [21,22,23,24].

## 2. Materials and Methods

The study comprises stratified data from four randomized controlled trials conducted in Norway, Sweden, and Denmark. The inclusion criteria were as follows: females with symptoms of UTIs, with the isolated causative agent *S. saprophyticus* (≥10^3^ CFU/mL), and allocated to pivmecillinam therapy. The exclusion criteria were participants with no follow-up data. For the detailed methodology from each study, please view articles [21,22,23] or the study protocols [25,26]. Antimicrobial susceptibility testing was performed by disk diffusion test methodology, according to EUCAST [27]. The presence of significant bacteriuria was defined as ≥10^3^ bacteria/mL for *S. saprophyticus* [28]. The MIC of mecillinam was determined in thirteen isolates from Study 4 [24], by E-test, according to the manufacturer (Biomerieux). The test was repeated three times for each isolate.

The inclusion criteria were similar in the four studies, namely: participants with uncomplicated lower urinary tract infections (acute uncomplicated cystitis) empirically treated with pivmecillinam. Several outcomes were also similar in the studies; the primary outcome was defined as follows:

A cumulative clinical effect—early clinical success, without recurrent UTIs at long-term follow-up.

The secondary outcomes were defined as follows:

An early clinical effect—clinically cured (symptom resolution) within eight days post-inclusion.

Long-term clinical effect—clinical status after about four weeks (i.e., a recurrence of UTI symptoms or not). Note, Study 3 [23] followed the participants for six months.

A bacteriological effect—defined as negative (insignificant growth or sterile urine) first-control-urine-sample (9–17 days post-inclusion).

Pivmecillinam was prescribed three times daily (t.i.d.) to all of the participants. The dosage and duration varied between the studies. In Study 1 [21] and 2 [22], pivmecillinam of 200 mg t.i.d. was given for three days (*n* = 45), but two participants in Study 1 received 10 days of therapy. In Study 3, pivmecillinam 400 mg t.i.d. was given for three days (*n* = 8). In study four, pivmecillinam 400 mg t.i.d. was given for five days (*n* = 13) or three days (added two days placebo; *n* = 6).

## 3. Results

Females (18–55 years) were treated with pivmecillinam for UTIs caused by *S. saphrophyticus* (*n* = 74; mean age: 25 years; standard deviation (SD) 5.8)—27 participants were treated with 400 mg t.i.d. (mean age: 25 years; SD 4.6) and 47 participants were treated with 200 mg t.i.d. (mean age: 24 years; SD 6.4). All of the case data are available in online (Appendix A, Appendix A).

For *S. saphrophyticus* UTIs, pivmecillinam´s cumulative clinical effect was 72%—53 out of 74 participants had clinical success with the treatment, and no recurrent UTIs in the follow-up periods (Table 1). The early clinical effect was 80%—59 out of 74 participants were clinically cured within eight days after starting treatment. The average time to symptom resolution was 3.1 days (SD 1.7). Six out of the 59 (10%) participants experienced clinical relapse within the long-term follow-up periods. The bacteriological effect was 86%—51 out of 59 participants had negative control urine.

In Study 1 [21], 26 out of 33 (79%) participants with *S. saphrophyticus* were clinically cured and none experienced recurrent UTIs. Of the 25 participants evaluable for a bacteriological effect, 22 (88%) achieved a bacteriological cure, and of the failures, one had a symptomatic relapse of *S. saprophyticus,* and in the other two, new bacteria were found in the control urine (only one was symptomatic). Of the two participants with 10 days of therapy, both achieved clinical cure and did not experience recurrent UTIs, but one had asymptomatic bacteriological relapse.

In Study 2 [22], 10 out of 12 (83%) participants with *S. saphrophyticus* were clinically cured, and three of these experienced recurrent UTIs. One person had clinical failure at day 8 and experienced clinical cure at day 14, without recurrent UTIs. Of the nine participants evaluable for a bacteriological effect, all experienced bacteriological cure.

In Study 3 [23], all eight participants with *S. saprophyticus* were clinically cured, but one experienced a symptomatic relapse, and 5 out of 7 (71%) achieved bacteriological cure.

In Study 4 [24], 13 out of 19 (68%) participants with *S. saprophyticus* were clinically cured, and 14 out of 16 (88%) achieved bacteriological cure. The failures were evenly distributed among the two treatment arms. Four participants received new antibiotic therapy (including the two with bacteriological failure). At long-term follow-up, 8 out of 10 (80%) participants had no relapse of symptoms, while two experienced recurrent UTIs—one was bacteriologically confirmed with a mecillinam susceptible *E. coli.* The MIC of mecillinam was determined for 12 *S. saprophyticus* isolates (i.e., 16 mg/L (*n* = 1); 32 mg/L (*n* = 4); 64 mg/L (*n* = 6); and 256 mg/L (*n* = 1)). Clinical success was seen in 7 out of these 12 participants (58%); the five participants who experienced failures were infected with isolates that had MICs of 32 mg/L (*n* = 3) and 64 mg/L (*n* = 2), respectively. A control urine sample was collected in 9 of these 12 cases, and all were negative.

## 4. Discussion

The bacteriological and clinical effects of pivmecillinam for UTIs caused by *S. saphrophyticus* were high in our studies.

Our findings are in concordance with Monsen et al. [19], and we agree with the authors that the breakpoints for *S. saphrophyticus* (especially in UTI) need to be revised, as pivmecillinam clearly has a high effect in most cases. Thulin et al. [29] have previously demonstrated that mecillinam-resistant *E. coli* reversed susceptibility ex-vivo in urine, and this is perhaps not only the case in acquired resistance, but also for bacteria that are considered to be “naturally resistant” to an antibiotic, such as *S. saprophyticus* and other Gram-positive bacteria for mecillinam.

Among the 12 cases with MIC determined in Study 4 [24], the MIC did not correlate with the outcome. Testing *S. saprophyticus* under more organ-specific conditions in vitro (e.g., in urine), may reveal other findings for mecillinam that better correlate with the outcome, compared to the standard testing, according to EUCAST. On the other hand, high urine concentrations of mecillinam might alone explain the beneficiary effect of the antibiotic against *S. saprophyticus,* as MICs seem to be well below the peak urine concentration. Future studies need to clarify the clinical breakpoints, and how susceptibility should be tested to best correlate with the clinical outcomes.

## 5. Conclusions

In conclusion, pivmecillinam is an excellent antibiotic for UTIs, and our studies found that the oral antibiotic clearly had high clinical and bacteriological efficacy in UTIs, even when caused by *Staphylococcus saprophyticus.*

## Figures and Tables

**Table 1 antibiotics-08-00057-t001:** Pivmecillinam efficacy in the treatment of urinary tract infections (UTIs) caused by *Staphylococcus saprophyticus.*

Study	Intervention		*n*	Clinical outcome			Bacteriological Outcome ^E^
	Dose ^A^	Duration		Early Cure (*n*) ^B^	Relapse (*n*) ^C^	Cumulative ^D^	
1 (reference 21)	200 mg	3 days	33	26	0	26/33	22/25 ^G^
		10 days	2	2	0	2/2	1/2 (new bacteria)
2 (reference 22)	200 mg	3 days	12	10	3	7/12	9/9
3 (reference 23)	400 mg	3 days	8	8	1	7/8	5/7 (relapse)
4 (reference 24)	400 mg	5 days	14	10	2	8/14	11/12 (relapse)
		3 days ^F^	5	3	0	3/5	3/4 (relapse)
**Collective outcomes:**			74	59 (80%)	6 (10%)	53/74 (72%)	51/59 (86%)

^A^ Thrice daily. ^B^ Resolution of symptoms during the first eight days. ^C^ No recurrent UTI at long-term (approximately four weeks) follow-up, with the exception of Study 4 (i.e., approximately six months follow-up). ^D^ Resolution of symptom during the first eight days, and no recurrent UTI in the long-term follow-up. ^E^ Sterile or no significant growth of bacteria in the control urine sample. ^F^ Added two days placebo. ^G^ One relapse and two new bacteria.

## Data Availability

All case data used in the analysis are available online with the article.

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
