# Peer review of "Pivmecillinam for Uncomplicated Lower Urinary Tract Infections Caused by Staphylococcus saprophyticus—Cumulative Observational Data from Four Recent Clinical Studies"

_antibiotics, 2019, doi:10.3390/antibiotics8020057_

Round 1

Reviewer 1 Report

Jansaker et al presents the results of four studies on the efficacy of pivmecillinam in treating lower urinary infections produced by S. saprophyticus

The scientific name of some bacterial species must be written in italic (lines 30, 31, 39, 86, 92, 102, 104, 108, 109, 114, 117, 123, 126, 129,130)

Latin statements such "bona fide" (line 37), "in vitro" (line 123) etc, should be written in italic

Line 39: after "Enterobacteriaceae", they should (could) add "sp".

Authors could write more about advantages of using pivmecillinam (resistance, side effects, etc)

Line 64: add "the" before "study four"

How did you ensure that the patients did not omit to administer some antibiotic doses?

It is mandatory to present your results in another manner to make them more explicit. For example in percents

Author Response

PEER-REVIEW nr 1

General response: Thank you for your peer-review and helpful comments.

Open Review

English language and style

( ) Extensive editing of English language and style required 
( ) Moderate English changes required 
( ) English language and style are fine/minor spell check required 
(x) I don't feel qualified to judge about the English language and style 

Yes

Can be improved

Must be improved

Not applicable

Does the introduction   provide sufficient background and include all relevant references?

( )

(x)

( )

( )

Is the research design   appropriate?

(x)

( )

( )

( )

Are the methods   adequately described?

(x)

( )

( )

( )

Are the results   clearly presented?

( )

( )

(x)

( )

Are the conclusions   supported by the results?

( )

(x)

( )

( )

Comments and Suggestions for Authors

Jansaker et al presents the results of four studies on the efficacy of pivmecillinam in treating lower urinary infections produced by S. saprophyticus

The scientific name of some bacterial species must be written in italic (lines 30, 31, 39, 86, 92, 102, 104, 108, 109, 114, 117, 123, 126, 129,130)

Response: Thank you for noticing this. In our file they are in italic. This must be a technical problem with the transformation of our document. We will make sure that before any possible publication that the final proof copy has all bacterial species in italic.

Latin statements such "bona fide" (line 37), "in vitro" (line 123) etc, should be written in italic

                      Response: As above.

Line 39: after "Enterobacteriaceae", they should (could) add "sp".

                      Response: Amended to: “…effect against several species of Enterobacteriaceae.”

Authors could write more about advantages of using pivmecillinam (resistance, side effects, etc)

Response: Amended. Several sections has been added to the introduction and marked in yellow.

Line 64: add "the" before "study four"

Response: We are not quite sure if this is most suitable in this sentence, but we added a “comma” to separate the specific details about the test from the first section of the sentence: “MIC of mecillinam was determined in thirteen isolates from study four [16], by E-test according…”

How did you ensure that the patients did not omit to administer some antibiotic doses?

Response: The adherence rates were high in each of the separate RCTs. The specific adherence rates can be viewed in each RCT article.

It is mandatory to present your results in another manner to make them more explicit. For example in percents

Response: Thank you for your comment. We have looked through the result section thoroughly. Unfortunately, we are not quite sure that we understand this comment. We have presented the results with percentages for each study respectively, and cumulatively for all studies. However, we have now moved the percentages up to immediately after the case data (eg. “26 out of 33 (79%”). Please, if we have misunderstood this comment; specify more and we will be glad to amend to it.

Reviewer 2 Report

The manuscript entitled "Pivmecillinam for uncomplicated lower urinary tract infections caused by Staphylococcus saprophyticus–cumulative observational data from four recent clinical studies," describe the outcomes of treatment of the lower urinary tract infection caused by S. saprophyticus, with a prodrug-pivmecillinam. 

 This reviewer has some minor questions/comments.

1. The organic structures of the pivmecillinam and mecillinam should be included in the paper.

2.  Why was pivmecillinam chosen over other drugs targeting gram +ve pathogens? 

3. In the case of study no. 4, did authors try synergy/combination therapy with different drugs for the cases where relapse/failure is the outcome?

4. The drug name and unit is not mentioned for MIC in SI table

5. In case study 4, why MIC against mecillinam was determined instead of pivmecillinam?

6. How three times daily conc./dose of pivmecillinam was determined?

7.  Line 124-126, the sentence ‘On the other hand, high urine….…S. saprophyticus” is not clear.

Author Response

PEER-REVIEW nr 2

General response: Thank you for your peer-review and helpful comments.

Open Review

English language and style

( ) Extensive editing of English language and style required 
( ) Moderate English changes required 
(x) English language and style are fine/minor spell check required 

Response: Thank you. Minor spell and grammatical corrections have now been done.
( ) I don't feel qualified to judge about the English language and style 

Yes

Can be improved

Must be improved

Not applicable

Does the introduction   provide sufficient background and include all relevant references?

( )

( )

(x)

( )

Is the research design   appropriate?

(x)

( )

( )

( )

Are the methods   adequately described?

( )

(x)

( )

( )

Are the results   clearly presented?

( )

(x)

( )

( )

Are the conclusions   supported by the results?

( )

(x)

( )

( )

Comments and Suggestions for Authors

The manuscript entitled "Pivmecillinam for uncomplicated lower urinary tract infections caused by Staphylococcus saprophyticus–cumulative observational data from four recent clinical studies," describe the outcomes of treatment of the lower urinary tract infection caused by S. saprophyticus, with a prodrug-pivmecillinam. 

 This reviewer has some minor questions/comments.

1. The organic structures of the pivmecillinam and mecillinam should be included in the paper.

Response: We have added information on the pharmacological properties in the introduction (marked with yellow).

2.  Why was pivmecillinam chosen over other drugs targeting gram +ve pathogens? 

Response: Treatment was initiated on symptoms alone (ie. empirically), and a urine sample was secured simultaneously.

3. In the case of study no. 4, did authors try synergy/combination therapy with different drugs for the cases where relapse/failure is the outcome?

Response: The primary physicians switched to another class of antibiotics. Synergism would be a good idea and something that is interesting to investigate in the future.

4. The drug name and unit is not mentioned for MIC in SI table

                      Response: Amended to MIC Mecillinam (mg/L)

5. In case study 4, why MIC against mecillinam was determined instead of pivmecillinam?

Response: Pivmecillinam (amdinocillin pivoxil) is the oral antimicrobial inactive prodrug of the antimicrobial active agent mecillinam (amdinocillin). Thus, MIC is determined for mecillinam since pivmecillinam lacks antimicrobial activity before the enzymatic hydrolysis of the pivoxil ester.

6. How three times daily conc./dose of pivmecillinam was determined?

Response: Three times daily is the customary dosing interval of pivmecillinam in Denmark and Norway. This is supported by theoretical considerations on PK/PD, based on PK data (Ref 5 in manuscript), clinical efficacy in ESBL (ref. 9-10 in manuscript) and E. coli susceptibility to mecillinam. Thus, the four randomized controlled trials used this interval.

7.  Line 124-126, the sentence ‘On the other hand, high urine….…S. saprophyticus” is not clear.

Response: The section in yellow was added to make this sentence clearer. On the other hand, high urine concentrations of mecillinam might alone explain the beneficiary effect of the antibiotic against S. saprophyticus, since MICs seem to be well below the peak urine concentration.